# Risk factors and transmission pathways associated with infant *Campylobacter spp.* prevalence and malnutrition: A formative study in rural Ethiopia

Sophie Budge[1☯], Megan Barnett[2☯], Paul Hutchings[1☯], Alison Parker[1☯*], Sean Tyrrel[1‡], Francis Hassard[1‡], Camila Garbutt[3‡], Mathewos Moges[4‡], Fitsume Woldemedhin[5], Mohammedyasin Jemal[5]

1 Cranfield University, Cranfield, United Kingdom, 2 British Geological Survey, Environmental Science Centre, Keyworth, Nottingham, United Kingdom, 3 People in Need UK, London, United Kingdom, 4 Department of Environmental Health, Hawassa University College of Medicine and Health Sciences, Hawassa, Ethiopia, 5 People In Need, Hawassa, Ethiopia

☯ These authors contributed equally to this work.
‡ These authors also contributed equally to this work.
* a.parker@cranfield.ac.uk

**Data Availability Statement:** All data created during this research is openly available from the

## Abstract

Early infection from enteropathogens is recognised as both a cause and effect of infant malnutrition. Specifically, evidence demonstrates associations between growth shortfalls and *Campylobacter* infection, endemic across low-income settings, with poultry a major source. Whilst improvements in water, sanitation and hygiene (WASH) should reduce pathogen transmission, interventions show inconsistent effects on infant health. This cross-sectional, formative study aimed to understand relationships between infant *Campylobacter* prevalence, malnutrition and associated risk factors, including domestic animal husbandry practices, in rural Ethiopia. Thirty-five households were visited in Sidama zone, Southern Nations, Nationalities and Peoples' region. Infant and poultry faeces and domestic floor surfaces (total = 102) were analysed for presumptive *Campylobacter* spp. using selective culture. Infant anthropometry and diarrhoeal prevalence, WASH facilities and animal husbandry data were collected. Of the infants, 14.3% were wasted, 31.4% stunted and 31.4% had recent diarrhoea. Presumptive *Campylobacter* spp. was isolated from 48.6% of infant, 68.6% of poultry and 65.6% of floor surface samples. Compared to non-wasted infants, wasted infants had an increased odds ratio (OR) of 1.41 for a *Campylobacter*-positive stool and 1.81 for diarrhoea. Positive infant stools showed a significant relationship with wasting (p = 0.026) but not stunting. Significant risk factors for a positive stool included keeping animals inside (p = 0.027, OR 3.5), owning cattle (p = 0.018, OR 6.5) and positive poultry faeces (p<0.001, OR 1.34). Positive floor samples showed a significant correlation with positive infant (p = 0.023), and positive poultry (p = 0.013, OR 2.68) stools. Ownership of improved WASH facilities was not correlated with lower odds of positive stools. This formative study shows a high prevalence of infants positive for *Campylobacter* in households with free-range animals. Findings reaffirm contaminated floors as an important pathway to

Cranfield Online Research Data at 10.17862/cranfield.rd.9907385.

**Funding:** SB is funded as a research student by People in Need and Cranfield University. CG, FW and MJ from People in Need advised on the practicalities of study design, helped with data collection and reviewed the manuscript. Project consumables and travel were funded by a GCRF QR grant awarded to Cranfield University.

**Competing interests:** The authors have declared that no competing interests exist.

infant pathogen ingestion and suggest that simply upgrading household WASH facilities will not reduce infection without addressing the burden of contamination from animals, alongside adequate separation in the home.

# 1. Introduction

## 1.1 Infant growth, infection and domestic animal exposure

Enteropathogen infection and associated diarrhoea in infancy and the relationship with linear growth failure (stunting) is a dynamic area of research in infant malnutrition. Whilst child deaths from diarrhoea dropped by over half in just 15 years between 2000–2015 [1], diarrhoeal episodes have not similarly decreased [2] suggesting a need for better measures to detect and prevent infection. Early diarrhoea and diarrhoea-related sequelae hold both acute and chronic consequences. Whilst good evidence indicates that a heavy early diarrhoeal burden does affect growth and worsen nutritional status [3–5], there is debate about its relative contribution to long-term growth faltering [6,7]. Other direct, biological causes under study include environmental enteric dysfunction (EED): a condition characterised by the disturbance of gut immunity, structure and function, which ultimately impairs nutrient absorption and linear growth–even without diarrhea [8–10]. Nonetheless, the common underlying factor to these different contributors is early exposure to pathogenic bacteria and repeated infection [11,12]. As such it is increasingly evident that stunting will not be resolved by improved nutritional intake or acute rehabilitation alone [13] but with parallel improvements in water quality, sanitation and hygiene (WASH) which act as a primary barrier to infection.

Recent cluster-randomised control trials have sought to investigate the effect of improved WASH, alone and in combination with nutrition supplementation, on child health. However different study designs and settings have for the most part failed to show consistent evidence for a reduction in diarrhoea or improvements in malnutrition indicators [14–19]. One possibility is that despite thorough design, interventions mainly focused on containing human excreta and did not consider (and conventionally have not considered) the role of animal faeces in domestic contamination and illness: [20] surprising given over 60% of infectious diseases in humans are caused by zoonotic pathogens [21]. Transmission pathways are not mutually exclusive, and inadequate separation of animals from the home environment may inevitably result in faecal-oral transmission through direct contact with animal faeces or contaminated soil, or faecal contamination of hands, food, objects or water sources [22–24]. Infants are also vulnerable to transmission routes specific to age-related behaviours, including contaminated floors, where they crawl and directly or indirectly ingest faecal material [25–27]. As such, animal faecal contamination is a neglected factor potentially contributing to infection, diarrhoea and linear growth failure.

## 1.2 Infant *Campylobacter* infection and transmission

Previous studies have sought to understand the disease burden attributed to animal faeces which acts as a transmission vector via the faecal-oral pathway [20–22]. Key zoonotic pathogens related to infant infection, growth failure and EED include *Giardia* [28,29], *enteroaggregative* and *enteropathogenic E. coli* [28–30], *Shigella* [31,32] and *Cryptosporidium* [33,34] which are transmitted across multiple pathways within the home and ingested through normal infant hand-to-mouth behavior [25,35]. Among those pathogens of highest concern,

*Campylobacter* consistently emerges as one of the key contributors to diarrhoea and malnutrition [31,32,34] and EED [28]. One of the most widespread infectious diseases, Campylobacteriosis is endemic across lower-income countries, especially in children [36]–responsible for 30,931 diarrhoeal-related deaths in 2015 [37]. The infectious dose for Campylobacteriosis is low compared to other bacterial infections, with reported minimum values of around 500 CFU leading to infection in adults: [38,39] this value may also be lower for infants where immune systems are immature. Infection is acute and generally self-limiting: however while mean excretion is reported at around seven days [36], the bacteria has been isolated from faeces up to two weeks following infection [40,41]. Prolonged excretion may enhance transmission and incidence [42] and where it also affects the epithelial barrier [43] may contribute to gut mucosal damage and other EED-like abnormalities [44].

Large studies across many different low-income settings have attributed both asymptomatic and symptomatic *Campylobacter* infection with shorter length attainment of up to one centimeter [32,44] and with changes in EED clinical markers [43,45]. Thermophilic *C. jejuni* (~90%) and *C. coli* are the most commonly isolated *Campylobacter* species in diarrhoeal disease [46], and as part of the normal intestinal flora of birds, poultry represents one of the major sources of transmission, contamination and infection [47]. An essential component of livelihoods and nutrition security, poultry ownership–particularly chickens–is ubiquitous across many low-income nations [48]. Largely free-ranging and dependent on scavenging, chickens frequently openly defecate inside the home and so infants are frequently exposed to, and often consume, chicken faeces and/or contaminated floor surface material during crawling or play [27,49,50]. As domestic floors are usually made of compacted soil, detection and removal of small poultry faeces is difficult and so Campylobacteriosis risk in crawling infants is high. Beyond six months of age critical developmental stages of weaning and crawling mean infection risk increases, [51] with obvious implications for short- and long-term growth and development. However, the evidence base describing the links between domestic animal ownership (particularly chickens), WASH facilities and use and infant nutritional status is limited to a few observational studies [27,52–55], which have not consistently measured *Campylobacter* carriage and/or infection. There is insufficient evidence to fully describe the extent to which infection is caused by exposure to domestic animals in low- and middle-income countries, and furthermore, if infant nutritional status affects whether infection is clinical or subclinical.

### 1.3 Study aims

In Ethiopia, despite substantial recent reductions, linear growth failure affected more than a third of infants in 2016 [56]. Ethiopia has one of the highest domestic animal densities per km$^2$ worldwide [57] and poultry are ubiquitous in rural households. Some research in Ethiopia has documented the proximity and exposure of infants to chickens and their faeces in regions [58] and the relationship with infant growth [54], and a few regional studies have associated *Campylobacter* infection with infant diarrhoea and malnutrition. [59–61]. However further research is required in Ethiopia on the epidemiology of infant *Campylobacter* prevalence and infant health outcomes and the relationship to poultry ownership and WASH facilities. Thus there is a need for further research which describes *Campylobacter* prevalence in young infants and the relationship to animal ownership and health outcomes, whilst also considering household WASH facilities and use. Further data is also needed on infection and age-related transmission pathways, including domestic floors which are of high risk to this age group [62–64]. This small study aimed to provide formative evidence toward the prevailing hypothesis that infant health is negatively associated with stools positive for *Campylobacter* and exposure to

domestic animals, whilst not mitigated by WASH facilities. It aimed to determine: i) Infant *Campylobacter* prevalence in a sample of rural, subsistence households in Sidama zone, Ethiopia with domestic animals ii) The relationship between both asymptomatic and symptomatic *Campylobacter* positive infants and anthropometric indices across households and iii) Risk factors and possible transmission pathways associated with infants positive for *Campylobacter*.

As this study was designed to provide formative evidence, a sample size calculation was not performed. Formative research is often conducted as part of the process of a larger study design and provides data for research teams to plan interventions or further data collection. Formative research is early phase data and is not powered to detect differences between groups. As such, this study results must be interpreted in this context, where it provided indicative data towards the hypothesis but was not sufficiently powered for conclusive evidence [65].

## 2. Methods

### 2.1 Country context and study sample

This small, formative study was conducted in the Southern Nations, Nationalities, and Peoples' region (SNNPR), Sidama zone (regional subdivision), Ethiopia, as the geographical outreach area of the non-governmental organisation People in Need. The study took part in the month of June 2019 –the start of the region's rainy season. Two rural kebeles (neighbourhoods) were chosen from a woreda (zonal subdivision) which remained representative of typical rural livelihoods across Sidama zone. A simple random sampling method was used to identify households fulfilling the eligibility criteria of having an infant aged 10−18 months and owning free-range poultry. The random sample is described as follows. After communication with a government Health Extension Worker (HEW) local to each kebele, the team produced a sampling frame for both kebeles of all infants aged 10−18 months from households known by the HEW to own poultry. For both sampling frames, households were sequentially numbered on paper and using a simple lottery method 17 and 18 infants were randomly drawn from the two kebele frames respectively for a total sample of 35 infants. Households were visited on a single occasion.

### 2.2 Survey and anthropometry

A survey previously validated in the region [50] assessed latrine type and use, handwashing practices and soap availability, domestic animal ownership and husbandry practices and infant diarrhoeal prevalence and duration. To assess diarrhoea, caregivers were asked the frequency of loose or watery stools during the last day and over the past seven days. World Health Organisation (WHO) criteria was applied retrospectively, where diarrhoea is defined as at least three loose or watery stools within a 24-hour period [66]. Reported diarrhoea was later compared with the quality of stool samples, where all cases of reported diarrhoea matched visible diarrhoeal stool consistency. Presence and evidence of use of a working latrine and handwashing station were also validated by direct observation. After primary introductions with the caregivers and informed consent, a fieldworker completed the survey with translation from the HEW. Anthropometry measures were infant recumbent length (measured to the nearest 0.1 cm) and weight (measured to the nearest 100 g), taken by trained personnel following standard procedures [67] using a hanging Salter scale and a portable, fixed base length board.

### 2.3 Sample collection and transport

A day prior to household visits, HEWs distributed sterile sample collection bags with a sterile scoop to households for faecal sample collection (Whirl-Pak[®] WPB01478WA, Sigma-Aldrich,

UK). Caregivers were shown how to use the sterile scoop and seal the bags to minimise contamination, and were requested to collect a fresh faecal sample from their poultry and infant as close as possible to sample collection within 24 hours. Households were instructed to collect poultry samples from inside the home. During the study visit a third sample was collected from the floor surface inside the home. The infant's mother was asked to indicate the location the infant usually plays, and a researcher collected a sample of compacted floor surface (approximately 20 g) into another collection bag. All samples were transported in an insulated cool bag on ice to the laboratory at Hawassa University College of Medicine and Health Sciences within five hours. Upon arrival to the laboratory, samples were stored refrigerated (2 −8°C) prior to analysis and plates were inoculated and incubated within two hours of arrival to the laboratory. Sample collection and transport methods echo similar methods in studies conducted in Ethiopia [64]. Thus each household sampling event (total = 35) comprised three samples (poultry and infant faeces and floor surface). Due to damaged collection bags, three floor surface samples were discarded to give a total of 102 samples analysed for presumptive *Campylobacter* spp. Samples were numbered anonymously which linked the relevant household but removed all identifiers.

## 2.4 Isolation of *Campylobacter* spp.

Presumptive thermotolerant *Campylobacter* spp. was isolated from fresh faecal samples from poultry, infants and floor surface samples. Methods were briefly as follows. Aseptic techniques were followed and samples weighed using sterile disposable weighing boats to $1 \pm 0.05$ g wet weight. Samples were then aliquoted into sterile plastic centrifuge tubes containing 9 mL of prepared sterile peptone water and vortexed well. For poultry faecal samples only, 100 μL of sample was pipetted into sterile tubes containing 900 μL of peptone water to prepare a 10-fold serial dilution up to $10^5$ dilution. 100 μL of floor surface and infant faecal samples and poultry faecal sample dilutions of orders $10^1$, $10^3$ and $10^5$ were drop plated on pre-labelled plates and spread using disposable L-shaped spreaders. Blood-free chromogenic CHROMagar™ Campylobacter media (CHROMagar™, France) was used for the selective detection and differentiation of presumptive thermotolerant *Campylobacter*, prepared and used according to manufacturer instructions [68]. Inoculated plates were allowed to dry under a laminar flow for approximately five minutes as per manufacturer instructions, inverted and stacked into anaerobic jars and incubated at 42°C for 48 hours under microaerophilic conditions. CampyGen™ 2.5 L sachets (Thermo Scientific™, UK) were used to obtain a hydrogen-free microaerophilic atmosphere of approximately 5% $O_2$, 10% $CO_2$ and 85% $N_2$, suitable for the growth of *Campylobacter* spp.

## 2.5 Identification of *Campylobacter* spp.

After 48 hours, presumptive *C. jejuni*, *C. coli* and *C. lari* appear on the chromogenic agar as intense red coloured colonies on a translucent base. Other non-target microorganisms are inhibited (i.e. small, blue colour or absent colonies [68]) and high specificity and sensitivity versus other media is well demonstrated [69–71]. Quality control and preparation of the medium was tested by isolating the ATCC® strain *C. jejuni* (33291) under representative conditions at Cranfield University prior to fieldwork. Blank samples with no growth confirmed no external contamination in all batches.

## 2.6 Ethics

At the start of each household visit, the study was introduced by the field team and HEW and informed consent was described to the caregiver in their first language of Amharic or Sidamo.

Fieldworkers tested the caregivers' understanding of consent by asking them questions regarding the study and the consent process, and explained all data was anonymised. As most adult caregivers were illiterate, oral consent and assent for their infant was recorded. The survey was written in English, translated to Amharic by the field team and verbally translated into Sidamo by a HEW. The study protocol was approved by two institutional review boards: Cranfield University Research Ethics Committee (CURES/7774/2019) and Hawassa University College of Medicine and Health Sciences (IRB/222/11).

## 2.7 Statistical analysis

Analyses were performed at the household level. Plates were visually inspected for presumptive *Campylobacter* spp. and recorded as growth or non-growth and the prevalence (as percentage) of positive poultry, infants and floor surfaces was calculated. Whilst the presence of *Campylobacter* does not necessarily indicate active infection, for the purpose of analysis, samples with presumptive *Campylobacter* growth were classified as 'positive' or with no growth as 'negative'. Positive infant faecal samples were then described as symptomatic (the positive stool sample was diarrhoeal), or asymptomatic (the stool sample was not diarrhoeal). Z scores were calculated for length-for-age and weight-for-length (LAZ and WLZ respectively) using the WHO 2006 Child Growth Standards [72]. Z-scores were categorised into stunting and wasting using the standard cut-off value less than −2 standard deviations of the reference [72]. Anonymised household survey data were entered into Microsoft Excel, coded for descriptive analysis and further analysed using SPSS (version 22.0, IBM, New York). Simple frequency distribution tests described survey response data, anthropometric data and *Campylobacter* prevalence. Fisher's exact test for independence tested associations between variables for the small sample size (5% significance). Results with significant p-values from the Fisher's exact test reported odds ratio (OR) risk estimates with corresponding 95% confidence intervals (CI).

## 3. Results

### 3.1 Survey and anthropometric data

Data were collected from all 35 households identified in the sampling frame. Results from the survey and anthropometric data are shown in Table 1. Average infant age was 15 months. Almost a third (31.4%) of infants had experienced diarrhoea within the past 7 days with an average duration of 3.1 days. Of households, 88.6% owned a latrine, most of which were improved pit latrines with a slab (82.9%). Less than half (40.0%) of households had some form of handwashing facility available (including a simple basin and jug) and half (51.4%) owned soap. Aside from poultry ownership, cattle was the second most common form of animal husbandry (total = 19, 54.3%). Regarding animal husbandry practices 97.1% of households reported that during the day their animals shared the same living space as the family, and 91.4% during the night. Mean WLZ score was -0.61 (range -2.14−0.64, SE 0.15) and mean LAZ score was -0.81 (range -2.53−0.94, SE 0.19). Overall, five infants (14.3%) were classified as wasted (WLZ <-2 SD), eleven (31.4%) as stunted (LAZ <-2 SD) and four infants both wasted and stunted (11.4%, WLZ and LAZ <-2 SD). Of those infants classified as wasted (total = 5), all had experienced diarrhoea within the past seven days (p<0.001; OR 1.83, 95% CI 1.07 −3.14). Diarrhoeal prevalence was not significantly related to stunting (p = 0.709).

### 3.2 *Campylobacter* prevalence and correlation with infant health measures

The following sections describe the relationships between survey variables, prevalence of presumptive *Campylobacter* and infant health outcomes. A total of 102 samples from poultry,

**Table 1. Infant and household characteristics (total = 35) including animal husbandry practices and anthropometric indicators, Sidama zone, Ethiopia.**

| Household characteristic | total (n) | Average or percent (%) of total |
|---|---|---|
| Infant sex | | |
| Male | 19 | 54.3% |
| Female | 16 | 45.7% |
| Average age (months) | | 15 |
| Diarrhoea during the last 7 days | 11 | 31.4% |
| Average duration of diarrhoea (days) | | 3.7 |
| Household owns a latrine | 31 | 88.6% |
| Household latrine type | | |
| Open defecation (no latrine) | 1 | 2.9% |
| Use neighbour's toilet (no latrine) | 3 | 8.6% |
| Pit latrine without slab | 2 | 5.7% |
| Pit latrine with slab | 29 | 82.9% |
| Household has a handwashing facility | 14 | 40.0% |
| Household has soap available | 18 | 51.4% |
| Household domestic livestock ownership | | |
| Chickens | 35 | 100% |
| Cattle | 19 | 54.3% |
| Goats | 11 | 31.4% |
| Donkey(s) | 2 | 5.7% |
| Livestock practices during the day: | | |
| Live outside | 35 | 100.0% |
| Live inside in the same room as the family | 34 | 97.1% |
| Live inside in a separate room to the family | 1 | 2.9% |
| Livestock practices during the night: | | |
| Live inside in the same room as the family | 32 | 91.4% |
| Live inside in a separate room to the family | 3 | 8.6% |
| Nutrition indicator | Total (n) | Percent (%) of total |
| Weight-for-length (WLZ) | | |
| -2 to -3 SD (wasted) | 5 | 14.3% |
| Length-for-age (LAZ) | | |
| -2 to -3 SD (stunted) | 11 | 31.4% |
| WLZ and LAZ | | |
| -2 to -3 SD (stunted and wasted) | 4 | 11.4% |

infants and floor surface were cultured for *Campylobacter* spp. Overall, *Campylobacter* was recovered from 48.6% (total = 17) of 35 infant faecal samples, 68.6% (total = 24) of 35 poultry faecal samples and 65.6% (total = 21) of 32 floor surface samples. Differences in the prevalence of positive samples which were symptomatic (a diarrhoeal stool sample) and asymptomatic (non-diarrhoeal stool, 'carriers') was seen among positive infants presenting with diarrhoeal stools (total = 10, 58.8%) versus without diarrhoea (total = 7, 41.2%) (p<0.001). Furthermore, infant who were wasted (low weight-for-length) versus not wasted were compared for *Campylobacter* prevalence. Those wasted were more likely to test positive for *Campylobacter* (p = 0.019; OR 1.41, 95% CI 1.04–1.92). Wasted infants thus appeared to have a 1.83 times the odds of diarrhoea and 1.41 times of a sample positive for *Campylobacter* versus those not wasted. However, diarrhoea was not associated with infant stunting (p = 0.709), nor was *Campylobacter* prevalence (p = 0.725).

### 3.3 Risk factors and transmission pathways related to infant *Campylobacter* prevalence

Further analysis using correlation explored the relationship between potential risk factors and transmission pathways to infant stools positive for *Campylobacter*. Considering associated risk factors, animal husbandry practices of keeping animals inside during the day and night (as a composite variable) was strongly correlated with increased odds of infants positive for *Campylobacter* (p = 0.027, OR 3.5, 95% CI 1.31–8.77). Owning donkeys or goats showed no association (p = 0.229 and p = 0.546 respectively), but owning cattle was significantly associated with increased odds, although with high uncertainty of effect (p = 0.018, OR 6.5, 95% CI 1.47 –28.90). Poultry faeces positive for *Campylobacter* showed significant correlation with infant *Campylobacter* (p<0.001, OR 1.34, 95%CI 1.21–1.69). However, owning a latrine, different types of latrine, owning a handwashing facility and ownership of soap were all not correlated (all p>0.5). Considering potential transmission pathways, positive floor samples showed a significant association, although again with high uncertainty of effect (p = 0.023, OR 7.0, 95% CI 1.5–23.4). Positive poultry faeces and positive floor samples were also highly correlated (p = 0.013; OR 2.68. 95% CI 1.64–12.62). The associations between risk factors and transmission pathways in relation to infant health outcomes are detailed in Table 2. Fig 1 also illustrates these associations whereby the dotted lines describe the main transmission pathways to an infant stool positive for *Campylobacter*.

## 4. Discussion

Results from this small cross-sectional study suggest that in these rural Sidamo households raising free-range domestic poultry, the prevalence of infants testing positive for *Campylobacter* spp. is high. With presumptive *Campylobacter* isolated in almost half of infant stools, results mirror high prevalence found in similar age infants in Zimbabwe (32.3%) [73], Mexico (66.0%) [42], Madagascar (43.3%) [74] and across eight low-resource settings where 84.9% of infants had at least one positive faecal sample by one year of age [45]. The high prevalence in this study may be due to sample collection during the rainy season where pooled water inside the home facilitates the spread of faecal bacteria: however other studies have found constant high prevalence not affected by seasonality [44,75]. In this study, 58.8% of the 17 infants positive for *Campylobacter* were symptomatic with diarrhoeal stools. With an average *Campylobacter* excretion of seven days [36] (and reported protracted excretion of more than 14 days [40]) this may lend support that current diarrhoea in these infants was from Campylobacteriosis. Studies in northern Ethiopia [59,60] and in the same zone as this study [61] suggest *Campylobacter* is a major regional cause of diarrhoea Comparing infants who were wasted (total = 5) versus those non-wasted, wasting was correlated with positive *Campylobacter* prevalence and diarrhoeal stools. Campylobacteriosis may have contributed to these outcomes, but it is likely other coexisting infections also contributed [32,76].

**Table 2. Odds ratios (OR) for exposure measures predicting an infant stool positive for *Campylobacter*, with corresponding confidence intervals (CI) and p values (significance < 0.05).**

| Variable | OR | 95% CI | P value |
|---|---|---|---|
| Infant wasting (WLZ <−2 SD) | 1.41 | 1.04–1.92 | 0.019 |
| Positive poultry faeces | 1.34 | 1.21–1.69 | <0.001 |
| Keeping animals inside (day and night) | 3.50 | 1.31–8.77 | 0.027 |
| Owning cattle | 6.50 | 1.47–28.90 | 0.018 |
| Positive floor sample | 7.00 | 1.50–23.40 | 0.023 |

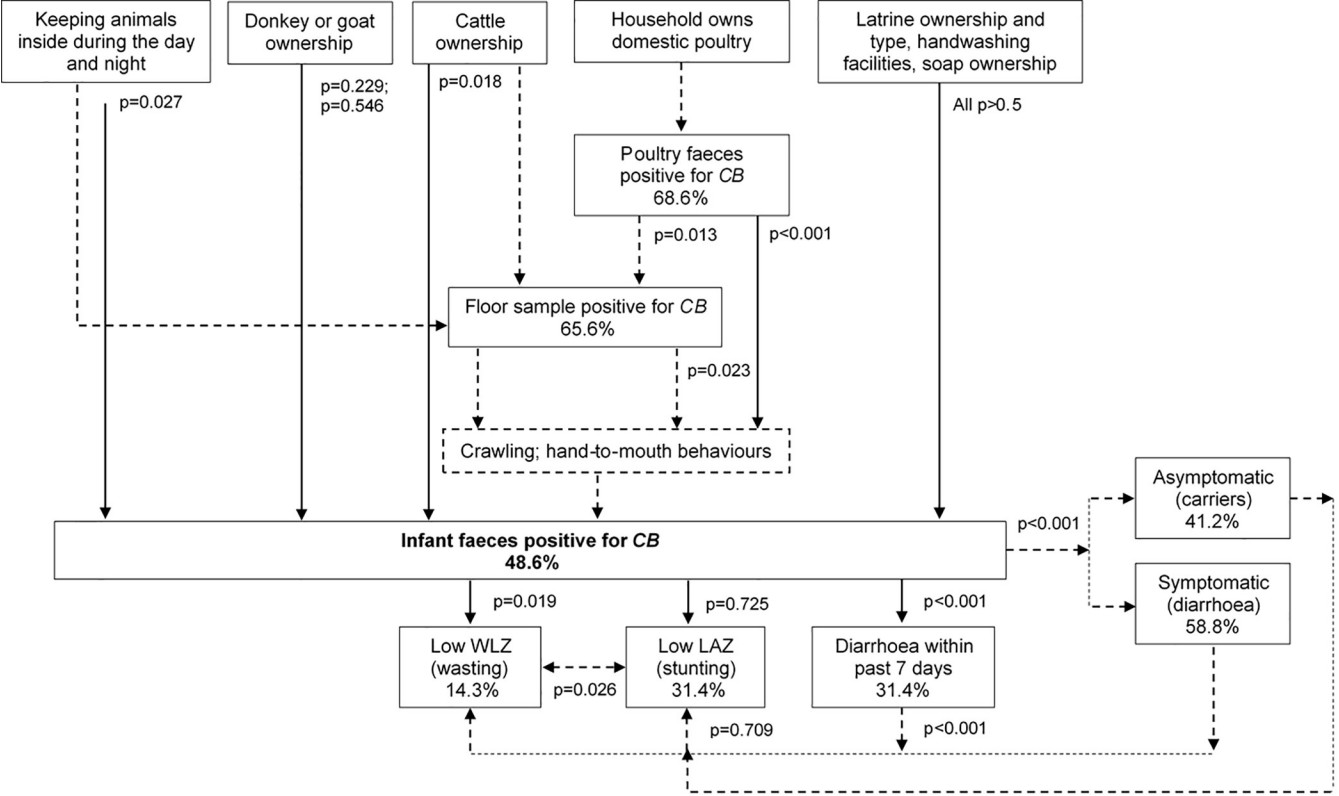

**Fig 1. Diagram exploring pathways between variables that predict infant stools positive for Campylobacter and the relationship with health outcomes.** Dotted lines demonstrate the hypothesised pathway linking poultry ownership, Campylobacter prevalence and health outcomes via clinical or subclinical disease. 'Symptomatic' infection refers to infants positive for Campylobacter who also had a diarrhoeal stool. P values <0.05 were deemed significant. This diagram is expanded in Fig 2. CB, Campylobacter; WLZ, weight-for-length; LAZ, length-for-age.

Whilst in early infancy infection may produce clinical symptoms and affect short-term weight, repeated enteropathogens colonisation may contribute to the development of EED. Although this study was not able to collect biological measures of EED, 41.2% (total = 7) of positive infant stools were asymptomatic (non-diarrhoeal stools). This supports findings from the MAL-ED study where subclinical infection was more strongly related to growth failure than overt diarrhoea [12]. Although positive stools showed no significant correlation with stunting, this may be partly due to the small sample size. Furthermore, research suggests that growth shortfalls resulting from early exposure to *Campylobacter* manifests later in infancy [31]. Studies have associated cohabiting with poultry with reduced length-for-age [53,77] and others have shown that infants who frequently test positive for *Campylobacter* have lower LAZ scores at 24 months of age, which had a stronger correlation with subclinical infection, or *Campylobacter* carriage [31,44]. Other studies have also demonstrated a relationship between poultry ownership and lower WAZ but not lower LAZ [53], suggesting both acute and chronic effects on health. Other significance lies in the overlap between wasting and stunting among infants in this group (p = 0.026), supporting evidence that the two forms of malnutrition can, and often do, coexist in the same infant [78], that they may share common causal factors of repeated carriage and/or infection [79].

This study aimed to further describe the relationship between domestic animal ownership and infant pathogen prevalence and growth, where free-roaming domestic animals may contribute to contamination of the home environment with pathogenic bacteria. Indeed in this

study, households were instructed to collect poultry samples from indoors and only two collected samples from outdoors, highlighting the ubiquity of poultry faeces inside the home. Infection is possibly transmitted to infants via age-specific behaviours and pathways. In this study, the significant risk factors that correlated with positive infant stools were specific animal husbandry practices of keeping animals inside during the day and night (ubiquitously in the same room as the family), owning cattle, positive domestic floor samples and positive poultry faeces. The analysis showed some uncertainty of effect and the small sample size may reduce the validity of findings, however the results do highlight specific risk factors to infants, including contaminated domestic floors as a potentially important transmission pathway. Longitudinal data from the MAL-ED team showed the effect of *Campylobacter* infection on growth is related to age–highlighting an increased level of risk as infants start to crawl [31]. Whilst this study did not capture hand-to-mouth contact events, previous research by this team in the same geographical area recorded infants mouthed their own hands or those of their caregiver a mean 31 and 21 times respectively over one hour, which were often visibly dirty (90.0% and 86.0% respectively) [50]. In the same study 35.0% of infants directly ingested floor surface material and poultry faeces was directly ingested by two infants (10.0%) [50]. Other studies have also recorded infants frequently ingesting poultry faeces from the floor during normal exploratory play [27,80,81].

Other factors not measured in this study, such as contaminated hands, food (particularly milk) and drinking water may account for the remaining sources of and transmission pathways to infant infection. Although a fastidious organism, *Campylobacter* is widespread in the environment, transmitted particularly through contaminated groundwater and stored drinking water [46], surviving for several days in an ambient environment [49]. Consequently research has suggested that in households where poultry are free-roaming, even with good water supply it is unlikely handwashing will effectively interrupt transmission [49]. *Campylobacter* transmission is also increased when WASH facilities are poor: [36] similar cross-sectional studies in Ethiopia also found higher *Campylobacter* prevalence in households without clean water and which had direct contact with chickens [82,83]. In this study latrine ownership and type (improved or not), ownership of handwashing facilities and soap were not correlated with stool samples negative for *Campylobacter*, perhaps suggesting that simply providing WASH facilities will not prevent transmission and infection. However it is possible facilities are also not used, particularly by children, which remains a limitation. In rural communities it can be difficult to assess and accurately report the use of latrines and soap for handwashing. Whilst in this study the visual inspection of latrines suggested they were all used, soap ownership would often be reported but not seen. Regardless, it seems logical that when sharing living spaces so closely, domestic animals contribute to infection from zoonoses and widespread contamination of multiple pathways. There are intrinsic and inseparable connections between these various transmission pathways. This is illustrated in Fig 2, which illustrates causal pathways to poor infant health outcomes when animal faecal contamination and age-specific infant behaviours are not considered as important risk factors.

The validity and broader applicability of findings from this study are mostly limited by the small sample size which may affect data validity and generalisability of the results. The single time point of testing in this formative research and the cross-sectional study design prevent determining causality. However, the results are emphasised as formative evidence, and support emerging hypotheses which associate free-range poultry ownership, household contamination and infant infection with undernutrition. Although this study intentionally sampled households who owned poultry, the risk of transmission may actually be greater than estimated as free-range chickens from neighbouring households may also increase contamination. Also, faecal samples from other domestic animals which also harbour *Campylobacter*, such as cattle

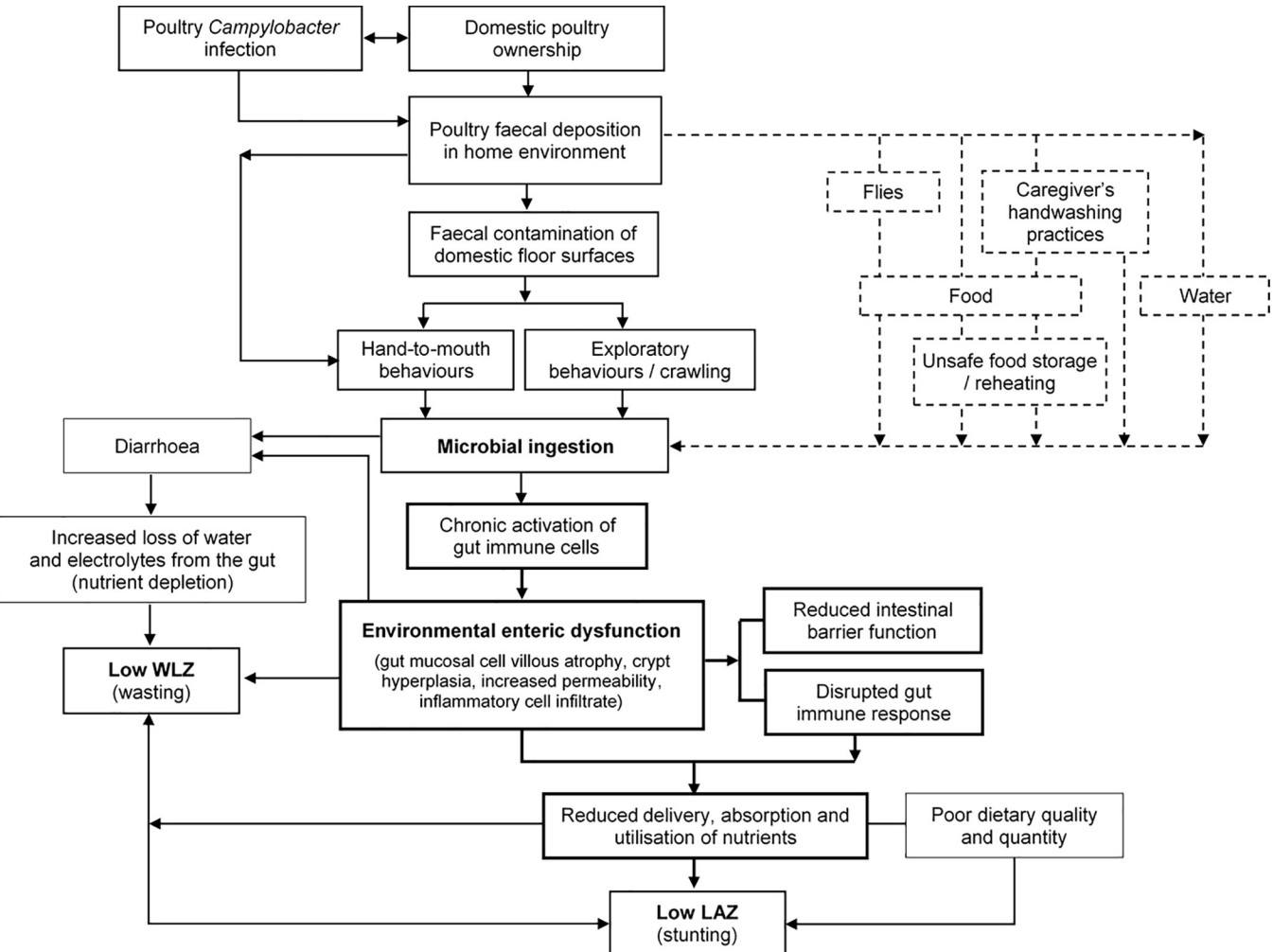

**Fig 2. The hypothesised pathways by which domestic poultry ownership contributes to acute and chronic infant malnutrition via infection from, and transmission of, Campylobacter.** The thicker part of the diagram illustrates the hypothesised relationship with environmental enteric dysfunction (EED). The dotted part of the diagram to the upper right constitutes the original 'F diagram', representing other transmission pathways by which infants are exposed to Campylobacter. Adapted alongside a previously published diagram [9] and the 'F published by Wagner, E. and Lanoix, J., 1958 [84].

[46], were not sampled. On the other hand there was no evaluation of the prevalence of other pathogenic or parasitic organisms, so it is not certain that the presumptive *Campylobacter* isolated in samples was the definite cause of wasting and/or diarrhoeal prevalence seen here. A few studies have reported mixed infections of *Campylobacter* and viral pathogens and their associations with infant morbidities [32,85]. This presence of *Campylobacter* alongside the carriage of multiple pathogens may correlate as a proxy for infants with greater overall levels of exposure to enteric pathogens in their environment; this in turn may associate with those with poor growth and/or wasting.

Lastly, the use of culture-based method alone holds limitations: firstly due to changes in *Campylobacter* cell physiology and loss of viability between sample deposition, collection, transport and plating (whereby cells enter the viable but non-cultivable [VBNC] state). This may have underestimated the true prevalence. On the other hand, culture holds limited sensitivity and high rates of false detection; [86] whilst there is evidence for good specificity of the agar in comparison and evaluation studies, there is no certainty of the rate of false positives in

this study. Lastly, whilst the culture media shows high specificity, it was not possible to differentiate between or quantify different *Campylobacter* species. The parallel use of qPCR alone or PCR with ELISA methods would enhance culture-based findings [31,87].

## 4.1 Conclusion

This formative study adds further preliminary evidence to the body of research documenting infant *Campylobacter* carriage and infection in households rearing free-range poultry. In these households, increased wasting and diarrhoea was seen in infants positive for presumptive *Campylobacter*. Repeated symptomatic infection and low weight may mean infants risk entering a spiral of weight loss and subsequent growth deficits. Alternatively, frequent carriage, or asymptomatic infection, and a high prevalence of stunting (although not correlated) suggest a longer-term impact of exposure to *Campylobacter* that may operate through EED. The time frame for when, and thresholds at which repeated *Campylobacter* infection becomes subclinical, contributes to the development of EED and affects growth are important remaining questions which a larger prospective cohort might address.

More broadly, this study also contributes to discussions around general WASH facilities and use, living conditions and the impact on reducing pathogen transmission. Where contaminated domestic floors are a risk factor for pathogen transmission to infants [59,88] and WASH facilities also appear have little effect in mitigating transmission, this emphasises the high thresholds of hygiene and living conditions necessary to improve infant health. While improvements to basic WASH usually included in interventions may address some secondary pathogen transmission routes, a remaining burden of infection may be expected when animals share the living space. An extensive, multifaceted approach to improve infant health will require not only improved WASH facilities, but working with communities to adapt current animal husbandry practices, encourage the safe handling and disposal of both animal and adult/infant faeces, safe preparation and storage of food, handwashing with soap after animal/faecal contact and education on the health risks of infant exposure. These multiple, concurrent needs form the rationale for the recent push toward 'transformative WASH' [89] or 'WASH++' [90]. Future research in the WASH sector must develop and test transformative WASH interventions if we are to achieve the high hygiene thresholds that support optimal infant growth.

## Supporting information

**S1 Data. STROBE checklist.**
(DOC)

## Acknowledgments

Sophie Budge wrote the manuscript. All other authors assisted in assessing both the paper quality and contributed to the writing and review of the manuscript. The authors wish to thank all of the People in Need team at the Hawassa office who assisted in data collection, logistics and planning. They would also like to thank Sara and Abebe at Hawassa University College of Medicine and Health Sciences and the wider faculty staff who gave their space and time. Finally the authors thank all of the study participants who kindly welcomed us into their homes and offered their time.

## Author Contributions

**Conceptualization:** Sophie Budge, Megan Barnett, Paul Hutchings, Alison Parker, Camila Garbutt.

**Formal analysis:** Sophie Budge.

**Funding acquisition:** Paul Hutchings.

**Investigation:** Sophie Budge, Megan Barnett, Mathewos Moges, Fitsume Woldemedhin, Mohammedyasin Jemal.

**Methodology:** Sophie Budge, Megan Barnett.

**Writing – original draft:** Sophie Budge.

**Writing – review & editing:** Megan Barnett, Paul Hutchings, Alison Parker, Sean Tyrrel, Francis Hassard, Camila Garbutt.

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
