## [Decision Letter · Decision Letter 0]

27 Jan 2020

PONE-D-19-33566

Domestic poultry ownership is associated with infant Campylobacter spp. infection and malnutrition: formative evidence from Ethiopia

PLOS ONE

Dear Dr Parker,

Thank you for submitting your manuscript to PLOS ONE. After careful consideration, we feel that it has merit but does not fully meet PLOS ONE’s publication criteria as it currently stands. Therefore, we invite you to submit a revised version of the manuscript that addresses the points raised during the review process.

We would appreciate receiving your revised manuscript by Mar 12 2020 11:59PM. To enhance the reproducibility of your results, we recommend that if applicable you deposit your laboratory protocols in protocols.io, where a protocol can be assigned its own identifier (DOI) such that it can be cited independently in the future. For instructions see: http://journals.plos.org/plosone/s/submission-guidelines#loc-laboratory-protocols

We look forward to receiving your revised manuscript.

Kind regards,

Yang Yang, Ph.D

Academic Editor

PLOS ONE

Additional Editor Comments:

This manuscript is well written. The study design has no major flaw, although one reviewer raised the question about the lab test. The presentation of the results is not well organized. The sample size is indeed too small, but this study still provides some significant results.

1. Table 1, I would suggest change the header “% of total” to “Average or Percent”, and move the values of “average age” and “average duration of diarrhea” from the “n” column to the “Average or Percent” column. Please also give the counts of diarrhea during last 7 days.

2. Page 13, lines 316-318. This sentence is very unclear to me. Are you comparing prevalence of infection between children with diarrhea during past 7 days and those without? If so, there should be an odds ratio. How are “symptomatic” vs “asymptomatic” carriers defined? Infections with diarrhea and infections without diarrhea in past 7 days?

3. Line 321, how was the OR of 1.41 calculated? If wasted kids were all infected, the odds of infection among wasted kids is then infinity, and the OR would be infinity too, as long as the infection rate among non-wasted children is not 100%. Given 95% CI for these ratios as well.

4. I would suggest making a table for all important OR estimates.

5. Figure 1, again, not sure what is the difference between “symptomatic (diarrhea)” and “Diarrhea within past 7 days”. One is 58.8% and the other is 31.4%, so they have to be different. Please give clear definition.

6. Were poultry feces collected indoor or outdoor? Were poultry feces common indoor?

3. We note that you have stated that you will provide repository information for your data at acceptance. Should your manuscript be accepted for publication, we will hold it until you provide the relevant accession numbers or DOIs necessary to access your data. If you wish to make changes to your Data Availability statement, please describe these changes in your cover letter and we will update your Data Availability statement to reflect the information you provide

4. Thank you for stating the following beneath the Acknowledgments Section of your manuscript:

'Sophie Budge is jointly funded as a research student by both Cranfield University and People in Need. This project was carried out with a grant from the 2018/19 Cranfield University QR Global Challenges Research Fund, supported by HEFCE/Research England. Further support was provided by the British Geological Survey and People in Need with logistical and project staff assistance.

'SB is partly funded as a research student by People in Need. https://www.clovekvtisni.cz/en/ CG, FW and MJ from People in Need adviced on the practicalities of study design, helped with data collection and reviewed the manuscript. '

Please provide an amended Funding Statement that declares *all* the funding or sources of support received during this specific study (whether external or internal to your organization) as detailed online in our guide for authors at http://journals.plos.org/plosone/s/submit-nowPlease state what role the funders took in the study.  If any authors received a salary from any of your funders, please state which authors and which funder. If the funders had no role, please state: "The funders had no role in study design, data collection and analysis, decision to publish, or preparation of the manuscript."

Reviewers' comments:

Reviewer's Responses to Questions

**Comments to the Author**

1. Is the manuscript technically sound, and do the data support the conclusions?

Reviewer #1: Partly

Reviewer #2: Yes

2. Has the statistical analysis been performed appropriately and rigorously? 

Reviewer #1: Yes

Reviewer #2: Yes

3. Have the authors made all data underlying the findings in their manuscript fully available?

Reviewer #1: Yes

Reviewer #2: Yes

4. Is the manuscript presented in an intelligible fashion and written in standard English?

Reviewer #1: Yes

Reviewer #2: Yes

5. Review Comments to the Author

Reviewer #1: The results described in this study represent a hot topic associated with important public health issues that need to be remediated as soon as possible. The manuscript was well written and I really appreciate the clarity and creativity of the figures, which really help in the interpretation of the data. Overall the study have great potential to help national and international institutes to fight Campylobacter transmission to infant and EED in Ethiopia; however major pitfalls remain in this study. For example, the suspected Campylobacter colonies isolated on CHROMagar were not confirmed via PCR due to technical difficulties; and the population samples used in this study is low (n=35 infants). Without confirmation of the taxonomic nature of the isolates by molecular and sequencing approach, I am highly suspecting that the Campylobacter prevalence reported in this study are highly overestimated. Further recovering Campylobacter from field samples can be very challenging in low income countries (I speak from experience); therefore, the use of a direct plating approach might not be the best strategy. This approach must be combined with more sensitive and specific methods.

Please find my comments and concerns below:

Abstract

• Missing word in the sentence line 25-26

• All numbers starting at the beginning of a sentence must be fully written (not numeric)

• Full name of SNNPR

• Please provide the r2 for the correlation data between the campylobacter prevalence and other variables studied (line 42)

• Only the use of microbiology approach (direct plating on selective medium) to identify Campylobacter is not good sufficient enough. Additional molecular approach (PCR) must be performed to confirm the isolates belong to the Campylobacter genus.

Introduction

• The introduction is too long

Materials and Methods

• Remove text from line 161 to 167.

• Line 173: Replace Zone by zone. Further, what do you mean by zone (=Woreda or kebele)?

• Line 173-175, More details are required concerning the simple random sampling method. Did the number of free-range chickens per household was recorded? Did the authors saw a correlation between the campylobacter prevalence or other recorded data and the number of free-range chickens.

• Line 178, “simple lottery method” Can you clarify?

• Line 180-182: I strongly disagree with the authors. The authors used these data to estimate the prevalence of Campylobacter in this specific area in Ethiopia. Therefore, the sampling method and the sample size are crucial in order to properly estimate the prevalence of Campylobacter. Please remove this sentence. Further, the authors must acknowledge that the population size low and restricted to only two kebeles. Please can you provide information concerning the expected statistical power of your analysis based population size. These information must be acknowledge in the manuscript as limitations of your study.

• Line 194-195, please can you explain how the comparison was performed. Did you have reference chart for the consistency (i.e. solid, pasty, liquid)?

• Line 202: what is the estimated time between the sampling and the incubation of the inoculated selective plates in microaerophilic conditions?

• Line 220: did the authors used a positive control in parallel during the processing the field samples (i.e. pure culture of C. jejuni or C. coli for example)?

• All numbers starting at the beginning of a sentence must be fully written (not numeric)

• Line 233: For how long the plates drying? This step is not necessary and might cause the death of Campylobacter.

• Line 233-237: I know by experience that this protocol is not sufficient to isolate Campylobacter from feces in Ethiopia due to the development of dense background (green colonies for example) and/or the presence of red campylobacter-like colonies that are not Campylobacter after confirmation via genus specific PCR (16S primer).

• Line 241-243: Unfortunately, this statement is not true. Our studies (unpublished data) showed that other genus can grow in this medium, which give campylobacter-like colonies. Due date, the only way we could differentiate them was via aerobic growth test or 16S sequencing.

• Line 249-252: I am scared that without PCR confirmation, you cannot affirm that your isolates are from the Campylobacter genus. This limitation must be acknowledged in your manuscript. I am suspecting that you are highly overestimating the Campylobacter prevalence based on your direct plating data. Our team faced the exact same issues in a previous project. Out of 1200 suspected Campylobacter isolates collected in Ethiopia using CHROMagar, less them 5% of the isolates were positive for Campylobacter via PCR. I would recommend buying external generators, which allow to continue your PCR cycles for 3-6 hrs.

• Line 273: you do not know whether Campylobacter or another microorganism of the gut microbiota is the causal agent of the diarrhea symptoms observed in the infant; thereby, you cannot use the term infected. It needs to be changed throughout the manuscript.

Results

• Line 311-323: You cannot use the term infected. It needs to be changed throughout the manuscript. All correlation data presented in the manuscript should be associated with a r2 value to fully appreciated the power of the correlation.

• Line 311: Did you observed difference in Campylobacter prevalence between the two kebeles. Why the authors did not present the CFU data from the direct plating?

• Line 341-342: This sentence is for the discussion, not the results

• Line 343-34: This sentence is for the fig legend, not the results

• Line 346: the legend in fig 1 and 2 should explain what the p values means. Further the quality of the figures is not good enough (too pixeled).

Discussion

• Line 364: the author must acknowledge the other potential reason of the diarrhea (other microorganism of the microbiota)

• Line 366-368: the population is too small, you cannot do such hypothesis.

• Line 375-377: your data do not support that given you don’t know if campylobacter is the causal agent of the diarrhea.

• Line 398-400: Again your population size is too small to make any hypothesis or conclusion. You may suggest a potential transmission from floor to infant but more studies must be done to confirm these preliminary results!

• Line 417-419; 453-455; 473-476: same comment then above. Your population size is too small to make any hypothesis or conclusion. These are preliminary results.

• The discussion is long and the authors miss to mention anything about the important limitation of this study: the population size and the lack of complete methodology to confirm the nature of the isolates.

Reviewer #2: Abstract:

Add specific stud area: SNNO has over 14,000,000 population. the sidama population is over 6 million

Add keyword at the end.

Any association between improved wash and children growth.

I donot think there is a author summery after abstract. If not remove it.

Introduction

Add local studies on prevalence of wash, campylobacter, and stunting in Ethiopia.

Methods:

Add detail methods of standardization and cite them

Is anthropometry is recommended for children 0-6 months, justify it or include in limitation of the study

Cite the acceptability of your sample collection and transport methods in low income setting.

Results and discussion:

I did not found the regression table in the section.

Add strength and limitation of this study)

6. PLOS authors have the option to publish the peer review history of their article (what does this mean?). If published, this will include your full peer review and any attached files.

Reviewer #1: Yes: Loic Deblais

Reviewer #2: Yes: Kedir Teji Roba

---

## [Author Response · Author response to Decision Letter 0]

18 Feb 2020

See detailed response attached.

---

## [Decision Letter · Decision Letter 1]

2 Apr 2020

PONE-D-19-33566R1

Risk factors and transmission pathways associated with infant Campylobacter spp. prevalence and malnutrition: A formative study in rural Ethiopia

PLOS ONE

Dear Dr. Parker,

Thank you for submitting your manuscript to PLOS ONE. After careful consideration, we feel that it has merit but does not fully meet PLOS ONE’s publication criteria as it currently stands. Therefore, we invite you to submit a revised version of the manuscript that addresses the points raised during the review process.

We would appreciate receiving your revised manuscript in 45 days. To enhance the reproducibility of your results, we recommend that if applicable you deposit your laboratory protocols in protocols.io, where a protocol can be assigned its own identifier (DOI) such that it can be cited independently in the future. For instructions see: http://journals.plos.org/plosone/s/submission-guidelines#loc-laboratory-protocols

We look forward to receiving your revised manuscript.

Kind regards,

Yang Yang, Ph.D

Academic Editor

PLOS ONE

Additional Editor Comments (if provided):

Please address the remaining questions of the reviewer. My questions have been addressed; however, I couldn't find figure legends in the revised manuscript.

Reviewers' comments:

Reviewer's Responses to Questions

**Comments to the Author**

1. If the authors have adequately addressed your comments raised in a previous round of review and you feel that this manuscript is now acceptable for publication, you may indicate that here to bypass the “Comments to the Author” section, enter your conflict of interest statement in the “Confidential to Editor” section, and submit your "Accept" recommendation.

Reviewer #1: All comments have been addressed

Reviewer #2: All comments have been addressed

2. Is the manuscript technically sound, and do the data support the conclusions?

Reviewer #1: Yes

Reviewer #2: Yes

3. Has the statistical analysis been performed appropriately and rigorously? 

Reviewer #1: Yes

Reviewer #2: (No Response)

4. Have the authors made all data underlying the findings in their manuscript fully available?

Reviewer #1: Yes

Reviewer #2: Yes

5. Is the manuscript presented in an intelligible fashion and written in standard English?

Reviewer #1: Yes

Reviewer #2: Yes

6. Review Comments to the Author

Reviewer #1: The results described in this study represent a hot topic associated with important public health issues that need to be remediated as soon as possible. The manuscript was well written and I really appreciate the clarity and creativity of the figures, which really help in the interpretation of the data. Overall the study have great potential to help national and international institutes to fight Campylobacter transmission to infant and EED in Ethiopia. The authors answered all the major concerns mentioned in the previous manuscript and properly edited the revised manuscript. Please find attached few minor comments

Reviewer #2: Risk factors and transmission pathways associated with infant Campylobacter spp. prevalence and malnutrition. is timely paper. and I would like to thank the authers.

All my previous comments were addressed. I have no any reservation the paper.

7. PLOS authors have the option to publish the peer review history of their article (what does this mean?). If published, this will include your full peer review and any attached files.

Reviewer #1: Yes: Loic Deblais

Reviewer #2: No

---

## [Author Response · Author response to Decision Letter 1]

9 Apr 2020

See response table in attachments

---

## [Editor Report · Decision Letter 2]

11 Apr 2020

PONE-D-19-33566R2

Risk factors and transmission pathways associated with infant Campylobacter spp. prevalence and malnutrition: A formative study in rural Ethiopia

PLOS ONE

Dear Dr. Parker,

Thank you for submitting your manuscript to PLOS ONE. After careful consideration, we feel that it has merit but does not fully meet PLOS ONE’s publication criteria as it currently stands. Therefore, we invite you to submit a revised version of the manuscript that addresses the points raised during the review process.

We would appreciate receiving your revised manuscript asap. To enhance the reproducibility of your results, we recommend that if applicable you deposit your laboratory protocols in protocols.io, where a protocol can be assigned its own identifier (DOI) such that it can be cited independently in the future. For instructions see: http://journals.plos.org/plosone/s/submission-guidelines#loc-laboratory-protocols

We look forward to receiving your revised manuscript.

Kind regards,

Yang Yang, Ph.D

Academic Editor

PLOS ONE

Additional Editor Comments (if provided):

I see some minor language issues with the authors' response to the minor editorial comments by one reviewer. Please fix them and return.
---

## [Author Response · Author response to Decision Letter 2]

15 Apr 2020

Two sentences have been rewritten where previously numbers had been written out in full at the start of the sentence.

---

## [Editor Report · Decision Letter 3]

17 Apr 2020

Risk factors and transmission pathways associated with infant Campylobacter spp. prevalence and malnutrition: A formative study in rural Ethiopia

PONE-D-19-33566R3

Dear Dr. Parker,

We are pleased to inform you that your manuscript has been judged scientifically suitable for publication and will be formally accepted for publication once it complies with all outstanding technical requirements.

With kind regards,

Yang Yang, Ph.D

Academic Editor

PLOS ONE
---

## [Editor Report · Acceptance letter]

28 Apr 2020

PONE-D-19-33566R3 

Risk factors and transmission pathways associated with infant *Campylobacter spp.* prevalence and malnutrition: A formative study in rural Ethiopia 

Dear Dr. Parker:

I am pleased to inform you that your manuscript has been deemed suitable for publication in PLOS ONE. Congratulations! Your manuscript is now with our production department. 

With kind regards,

on behalf of

Dr. Yang Yang 

Academic Editor

PLOS ONE